# Carnosic Acid Encapsulated in Albumin Nanoparticles Induces Apoptosis in Breast and Colorectal Cancer Cells

**DOI:** 10.3390/molecules27134102

**Published:** 2022-06-25

**Authors:** Katren F. Khella, Ahmed I. Abd El Maksoud, Amr Hassan, Shaimaa E. Abdel-Ghany, Rafaat M. Elsanhoty, Mohammed Abdullah Aladhadh, Mohamed A. Abdel-Hakeem

**Affiliations:** 1Department of Pharmaceutical Biotechnology, College of Biotechnology, Misr University of Science and Technology, Giza 3236101, Egypt; katren.farg2014183@gmail.com (K.F.K.); ahmed.ibrahim@gebri.usc.edu.eg (A.I.A.E.M.); mohamed.abdelhakim@must.edu.eg (M.A.A.-H.); 2Department of Industrial Biotechnology, Genetic Engineering and Biotechnology Research Institute (GEBRI), University of Sadat City, Sadat City 32897, Egypt; rafaat.elsanhoty@gerbi.usc.edu.eg; 3Department of Bioinformatics, Genetic Engineering and Biotechnology Research Institute (GEBRI), University of Sadat City, Sadat City 32897, Egypt; 4Department of Environmental Biotechnology, College of Biotechnology, Misr University of Science and Technology, Giza 3236101, Egypt; shaimaa.ibraheem@must.edu.eg; 5Department of Food Science and Human Nutrition, College of Agriculture and Veterinary Medicine, Qassim University, Buraydah 51452, Saudi Arabia

**Keywords:** carnosic acid (CA), bovine serum albumin nanoparticles, BCL-2, COX-2, GCLC

## Abstract

Carnosic acid (CA) is a natural phenolic compound with several biomedical actions. This work was performed to study the use of CA-loaded polymeric nanoparticles to improve the antitumor activity of breast cancer cells (MCF-7) and colon cancer cells (Caco-2). CA was encapsulated in bovine serum albumin (BSA), chitosan (CH), and cellulose (CL) nanoparticles. The CA-loaded BSA nanoparticles (CA-BSA-NPs) revealed the most promising formula as it showed good loading capacity and the best release rate profile as the drug reached 80% after 10 h. The physicochemical characterization of the CA-BSA-NPs and empty carrier (BSA-NPs) was performed by the particle size distribution analysis, transmission electron microscopy (TEM), and zeta potential. The antitumor activity of the CA-BSA-NPs was evaluated by measuring cell viability, apoptosis rate, and gene expression of GCLC, COX-2, and BCL-2 in MCF-7 and Caco-2. The cytotoxicity assay (MTT) showed elevated antitumor activity of CA-BSA-NPs against MCF-7 and Caco-2 compared to free CA and BSA-NPs. Moreover, apoptosis test data showed an arrest of the Caco-2 cells at G_2_/M (10.84%) and the MCF-7 cells at G2/M (4.73%) in the CA-BSA-NPs treatment. RT-PCR-based gene expression analysis showed an upregulation of the *GCLC* gene and downregulation of the BCL-2 and COX-2 genes in cells treated with CA-BSA-NPs compared to untreated cells. In conclusion, CA-BSA-NPs has been introduced as a promising formula for treating breast and colorectal cancer.

## 1. Introduction

Breast and colon cancer is a primary cause of malignant tumors and one of the most important causes of disability and death globally [1]. The disease is a typical emergency that requires continuous work for new treatment methods [2].

Natural products have recently attracted more interest due to their potential pharmacological properties and lower toxicity for synthesizing effective drugs [3]. Carnosic acid (Figure 1) (MW~330 Da) is a phenolic diterpene derived from *Rosmarinus officinalis* [4]. It has broad pharmacological properties involving antitumor, antiviral, and anti-inflammatory activities [5,6]. However, low water solubility and poor bioavailability of CA limit its in vivo anticancer effects [7].

Nanoparticle-based drug delivery systems can enhance the bioavailability and antitumor activity of chemotherapeutic drugs [8]. Also, these formulations may change the biodistribution of the drugs, reduce drug resistance, diminish nonspecific toxicity and protect the drugs from enzymatic degradation [9,10]. Polymeric nanoparticles have been developed for various drug-delivery applications due to their biocompatibility, simplicity, and low cost of fabrication [11,12]. Currently, nanomedicine plays a central role in biomedical applications, such as diagnostic and therapeutic applications [12].

Albumin is an attractive macromolecule carrier that can be obtained from a various sources, including egg white (ovalbumin), bovine serum albumin, and human serum albumin (HSA). It is a water soluble protein that maintains the osmotic pressure, binding, and transport of nutrients to the cells [13]. It can be used as an eco-friendly biomaterial in drug delivery because of its biocompatibility and easy degradation without toxicity. Bovine serum albumin (BSA) micelles have been developed to improve the bioavailability of these drugs and reduce their toxicity [14]. Histological observations demonstrated that bovine albumin has no adverse effects following frequent administration by the intranasal route [15]. Additionally, the application of BSA as a drug delivery agent was shown in an MCF-7 xenograft mouse model where the in vivo antitumor evaluation of FA-Rg5-BSA NPs were shown to be more effective in inhibiting tumor growth than Rg5 [16,17].

The activity of γ-glutamylcysteine synthetase is associated with elevated GSH levels in various cancer types [18]. Glutamate cystyl ligase catalyzed subunit (GCLC) is an essential enzyme involved in GSH biosynthesis and has been mentioned to be abnormally expressed in tumor tissue [19]. In colorectal cancer, GCLC has been shown to overexpress liver metastases and encourage cancer cell survival. Also, many reports have shown that GCLC activation is related to antitumor drug resistance in breast cancer [20,21].

Cyclooxygenase-2 (COX-2) is an inducible enzyme that catalyzes the synthesis of prostanoids, including prostaglandin, which is considered a significant mediator of inflammation and angiogenesis [22]. Moreover, COX-2 is overexpressed in cancer cells which causes progressive tumor growth and resistance of those cells to conventional therapy [23]. The Bcl-2 protein, encoded by the Bcl-2 gene, plays an anti-apoptotic role and inhibits the programmed [24]. The impact of CA on the expression of GCLC, COX-2, and BCL-2 may clear the mechanism of its antitumor activity. This work intended to study the anticancer molecular mechanisms of CA loaded in polymeric nanoparticles in breast cancer (MCF-7) and colon cancer (Caco-2) cell lines.

For this purpose, carnosic acid was encapsulated in different polymeric nanoparticles, namely chitosan (CH), bovine serum albumin (BSA), and cellulose (CL). The prepared nanoformulations were characterized to select the best formula. The selected formula was utilized as a treatment for MCF-7 and Caco-2. The antitumor activity was followed via MTT assay and cell cycle analysis. Moreover, GCLC, BCL-2, and COX-2 gene expressions were also evaluated before and after treatment.

## 2. Materials and Methods

### 2.1. Materials

Breast cancer cells (MCF-7) and colorectal cancer (Caco-2) were purchased from VACSERA, Giza, Egypt. Carnosic acid standard (purity 99%), chitosan (Deacetylation degree 95%, molecular weight 80 kDa), and BSA were purchased from Sigma Aldrich. Ethyl acetate and hexane were purchased from Fisher Scientific. Sodium tripolyphosphate (TPP) and glutaraldehyde were purchased from Merck. All the other chemicals and reagents were analytical grade.

### 2.2. Preparation of Nanoparticles

#### 2.2.1. Synthesis of CA-Loaded Albumin Nanoparticles

The preparation of carnosic acid-loaded albumin nanoparticles was carried out according to the previous report [25]. Typically, 100 mg of BSA was dissolved in 10 mL of deionized water containing 10 mg of carnosic acid. The dissolved albumin was precipitated as NPs by adding 40 mL of ethanol. Then, 16 mM 8% glutaraldehyde was added for crosslinking of precipitated protein. The obtained CA-BSA-NPs were collected by centrifugation at 12,000 rpm for 20 min. The empty carrier was obtained by the same method without adding carnosic acid.

#### 2.2.2. Synthesis of CA-Loaded Chitosan Nanoparticles

Chitosan nanoparticles loaded with carnosic acid (CA-CH-NPs) were prepared by the ionic gelation method [26]. Initially, 100 mg of chitosan was dissolved in 2% aqueous acetic acid (20 mL), and then 10 mg of carnosic acid was added. The 0.1% (*w*/*v*) polyanionic solution was prepared by dissolving TPP in deionized water. The nanoparticles are formed by mixing the two solutions by dropping the TPP into the chitosan solution with continuous stirring. Then, the nanoparticles were centrifuged at 10,000 rpm for 15 min, washed with distilled water, and lyophilized. Similarly, empty chitosan nanoparticles were prepared like the previous method without adding carnosic acid.

#### 2.2.3. Preparation of CA-Loaded Cellulose Nanoparticles

The cellulose extraction from *Chlorella vulgaris* algae was described by Hamouda et al. 2021 [27]. In brief, Five grams of dried *Chlorella vulgaris* algae powder were extracted with 68 mL of toluene and 32 mL of ethanol. The residue was collected by filtration, suspended in 100 mL of 4% aqueous NaOH, and heated for two hours at 80 °C. After washing with distilled water, the pellets were heated for two hours at 70 °C in 100 mL of 10% sodium hypochlorite pH 4.8. Finally, the residue was heated in sulfuric acid (65% *wt*/*v*) at 45 °C for 45 min. Pure cellulose was obtained after dialysis using distilled water until the pH became 7.

Preparation of cellulose nanoparticles loaded with carnosic acid (CA-CL-NPs) was performed according to the previously described method [28]. Eight milligrams of CA dissolved in 50 mL of acetone was gradually added to CL (40 mg) water suspension. The mixture was allowed to be stirred overnight. CA-CL-NPs were collected by centrifugation at 10,000 rpm for 10 min, freeze-dried, and stored at 4 °C.

#### 2.2.4. Evaluation of Drug Loaded Efficiency

Carnosic acid encapsulation efficiency (EE%) and loading capacity (LC%) were calculated according to Equations (1) and (2) [29]. Briefly, the formed nanoparticles (CA-CH-NPs, CA-BSA-NPs, and CA-CL-NPs) were separated from the aqueous medium containing the free drug by centrifugation at 9000 rpm for 30 min. The amount of remaining CA was estimated by measuring the absorbance at 298 nm using a UV spectrophotometer (Jenway 6305, Staffordshire, UK).
(1)$$EE\left(\%\right)=\frac{Initial\;amount\;of\;CA-Remaning\;amount\;of\;CA}{Initial\;amount\;of\;CA}\times 100\;$$
(2)$$Loading\;Capacity\;\left(\%\right)=\frac{Initial\;amount\;of\;CA-Remaning\;amount\;of\;CA}{weight\;of\;dried\;NPs\;}\times 100$$

#### 2.2.5. In Vitro Drug Release Study

Three release media were prepared as follows: KCl/HCl buffer at pH 1.5 containing 30% ethanol (medium 1), phosphate buffer at pH 7.2 containing 30% ethanol (medium 2), phosphate buffer at pH 7.9 containing 30% ethanol (medium 3). The release profile of loaded CA was studied for all formulations (CA-CH-NPs, CA-BSA-NPs, and CA-CL-NPs) in the release media. Typically, 200 mg of each formulation was suspended in 2 mL of distilled water and placed in a dialysis bag (Cutoff 70-100KD), and the bag was immersed in 15 mL of release medium 1 (pH 1.5) and maintained under 37 °C and 100 rpm. After two hours, the dialysis bag was transferred to medium 2 (pH 7.2) for 6 h and finally to medium 3 (pH 7.9) for 2 h. At certain time intervals (t), 1 mL of releasing medium was substituted with 1 mL of fresh medium. The extracted drug was measured spectrophotometrically at λ = 298 nm [30]. The results were plotted as cumulative released percent versus time according to the following equation.
$$Cumulative\;release\;percentage=\overset{t}{\underset{t=0}{\sum }}\frac{CA\left(t\right)}{CA\left(i\right)}\times 100$$
where *CA*(*i*) and *CA*(*t*) are the initial concentration of *CA* and at a specific time, respectively.

The best formula (highest EE%, LC%, and in vitro release rate) was selected and used for further investigations.

#### 2.2.6. Nanoparticle Characterization

The morphology of BSA-NPs and CA-BSA-NPs was investigated by transmission electron microscopy (TEM) (JEOL, JAM-2100-HR-EM). The hydrodynamic size and zeta potential were measured by dynamic light scattering (Nicomp Nano Z3000 Zeta Potential Analyzer).

#### 2.2.7. Cell Lines

The Dulbecco’s Modified Eagle’s Medium (DMEM), and RPMI 1640 medium containing 10% fetal calf serum, 100 U/mL of penicillin, and 100 μg/mL of streptomycin were used to culture the MCF-7 and Caco-2 cells. The cells were maintained at 37 °C in a humidified incubator containing CO_2_ 5% (*v*/*v*). At 85% confluence, the cells were dissociated using trypsin (0.25% *w*/*v*) and then sub-cultured into 75 cm^2^ flasks and six-or 96-well plates (TPP-Swiss) depending on the experiments [31].

#### 2.2.8. Cell Viability Test

The MTT assay was used to determine the viability of MCF-7 and Caco-2 cells [32]. Briefly, in 96-well plates, 1 × 10^5^ cells/well were seeded and exposed to CA, BSA-NPs, and CA-BSA-NPs at different concentrations (100 μg/mL, 50 μg/mL, 25 μg/mL, 12.5 μg/mL, and 6.25 μg/mL).After exposure, for 24 h, the culture medium was decanted, and the plates were washed using phosphate-buffered saline (PBS) pH 7.2 ± 0.2. After adding the MTT solution (0.5 mg/mL of PBS) to the treated/untreated cells, the plates were incubated for 4 h at 37 °C. One hundred microliters of DMSO was added for 10 μL of MTT. The absorbance was measured at a wavelength of 570 nm using an ELISA reader apparatus (ELX-800n, Biotek, Winooski, VT, USA). The cell viability for treated cells was represented as a percentage to control cells. The IC_50_ values of CA, BSA-NPs, and CA-BSA-NPs at 24 h was calculated using the Microcal Origin 6.0 Professional analysis software and used for all subsequent assays.

#### 2.2.9. Cell Cycle Arrest

A specific culturing media containing CA-BSA-NPs, 2.60 μg/mL for Caco-2 cells and 6.02 μg/mL for MCF-7 cells was prepared. The cells (3 × 10^5^ cells/well) were seeded and cultured for 24 h. The cells were fixed overnight at −4 °C in ethanol 75% and then incubated in the dark with PI staining solution (50 ng/mL) and RNase A (0.1 mg/mL) for 15 min. Flow cytometry (BD FACSCalibur-USA) was used to determine the DNA content of the cells [31,33].

#### 2.2.10. Cell Apoptosis Assay

The Annexin V-FITC/PI apoptosis staining was performed by the Annexin V-FITC apoptosis staining kit (Annexin V-FITC-BD Bioscience Pharmingen^TM^, San Diego, CA, USA) [32]. In brief, the Caco-2 and MCF-7 cells were cultured and treated as mentioned above (Section 2.2.9). The cells were collected and incubated in a mixture of 100 μL of 1X Binding Buffer002Cx and 100 μL of Annexin V. Annexin V-FITC (5 μL) and propidium iodide (5 μL) were incubated at room temperature for 15 min in the dark, then added to 400 μL of 1X binding buffer and processed by flow cytometry within one hour for maximal signal. The cells were examined by flow cytometry (BD FACSCalibur-USA).

#### 2.2.11. Gene Expression by Real-Time PCR

As mentioned in Section 2.2.9, the Caco-2 and MCF-7 cells were incubated with the proper concentration of CA-BSA-NPs. Primer designs were performed by the Primer-BLAST tool, NCBI (Table 1). Total RNAs were extracted from untreated and treated cells using the RNeasy mini-Kit (Qiagen, Valencia, CA, USA CAT. No. (EN0525)) according to the manufacturer’s instructions.

The quantitative assessment of COX-2, GCLC, and BCL-2 gene expression, was carried out using StepOne Plus thermal cycler (Applied Biosystems, Warrington, UK) according to the following procedure. Firstly, the cDNA synthesis was performed using a High-Capacity cDNA Reverse Transcriptase kit (Applied Biosystems, Waltham, MA, USA). After that, the cDNA was amplified with the Syber Green I PCR Master Kit (Fermentas) using the Step One instrument (Applied Biosystems, USA CAT. NO. (4368814)) as follows: 10 min at 95 °C for enzyme activation, followed by amplification step: 40 cycles of 15 s at 95 °C, 20 s at 55 °C, and 30 s at 72 °C. The target gene’s expression changes were normalized relative to the mean critical threshold (CT) values of β-actin as a housekeeping gene by the ΔCt method.

## 3. Results

### 3.1. Encapsulation Efficiency and Loading Capacity

According to equation 1, the encapsulation efficiencies of CA-CL-NPs, CA-BSA-NPs, and CA-CH-NPs were 22.40%, 40.80%, and 42.43%, respectively, while the loading capacities were 4.16%, 8.02%, and 8.34%, respectively.

### 3.2. In-Vitro Drug Release Study

In the current study, we assessed the release profile of carnosic acid from albumin, chitosan, and cellulose nanoparticles at three release media with pH values of 1.5, 7.2, and 7.9, respectively. The release of CA from CA-CH-NPs starts after 30 min with approximately 10.3% to 11.5% at 1.5 pH and 16.1 to 18.7% at 7.2 and 7.9. Release of CA from CA-CL-NPs was approximately 16–30% at 1.5 pH, 37–59% at 7.2 pH, and 60% at 7.9 pH. Finally, the release of CA from CA-BSA-NPs was the greatest; it was approximately 39% at 1.5 pH, 75% at 7.2 pH, and 80% at 7.9 pH (Figure 2).

### 3.3. Characterization of Carnosic Acid Load on BSA-NPs

The TEM images showed spherical shape particles of BSA-NPs and CA-BSA-NPs with a size range from 39.35–59.78 nm and 97.29–144.26 nm, respectively (Figure 3A,B). Moreover, BSA-NPs revealed hydrodynamic sizes ranging from 208 to 604 nm, with a prominent peak at 291 nm, polydispersity index (PDI) of 0.29, and zeta potential of −29.20 mV (Figure 4A,B). CA-BSA-NPs had a hydrodynamic size range of 330 to 662 nm after CA loading, with a primary peak at 520 nm, polydispersity index (PDI) of 0.119, and zeta potential of −21.03 mV (Figure 5A,B).

### 3.4. Cell Viability

The IC_50_ values of free CA against Caco-2 and MCF-7 cells were 8.29 μg/mL and 27.43 mg/mL, respectively. Interestingly, the CA-BSA-NPs significantly reduced the IC_50_ values to 2.60 and 6.02 μg/mL for Caco-2 and MCF-7, respectively. Furthermore, after treatment with the same concentration, the MTT assay showed a significant decrease in the cell viability after treatment with CA-BSA-NPs, while the BSA-NPs and free CA showed nonsignificant cytotoxicity (Figure 6A,B).

### 3.5. DNA Content Analysis

The flow cytometry was used to evaluate the effect of CA-BSA-NPs on the distribution of cell cycle phases. Results showed that CA-BSA-NPs cell growth arrest was at G_2_/M with 32.75% for Caco-2 cancer cells compared to control, 11.87%, as shown in Figure 7A,B. Also, CA-BSA-NPs cell growth arrest was at G_2_/M with MCF7 cancer cells compared to control, 13.49% for MCF7 cancer cells, as shown in Figure 8A,B.

### 3.6. Cell Apoptosis Assay

Annexin V-FITC/PI staining was used to identify apoptosis in the treated and untreated cancer cell lines. Data in Figure 9A showed that total apoptosis (undergone early and late apoptosis) in untreated cells was 0.64%, while CA-BSA-NPs-treated Caco-2 cancer cells yielded 17.74% (Figure 9B). On the other hand, CA-BSA-NPs had a lower effect on MCF-7 cells. As Figure 10B displayed, the percentages of apoptotic cells (including early and late apoptotic cells) in MCF-7-treated cells were increased to 10.45% compared with control (1.05%) in Figure 10A. As a result, the data displayed that CA-BSA-NPs induce apoptosis on both MCF-7 and Caco-2 cell lines.

### 3.7. Gene Expression

The current study evaluated the expression of BCL-2, COX-2, and GCLC genes in Caco-2 and MCF-7 cells (Figure 11A–C). Results revealed that in MCF-7, the treatment of CA-BSA-NPs (6.02 μg/mL) has significantly downregulated BCL-2 and COX-2 (FC  =  0.469 and 0.29, respectively), while the expression level of GCLC was significantly increased (FC  =  3.7) (Figure 11A,C).

In Caco-2, the GCLC gene was also significantly upregulated (FC  =  2.03), while the expression of BCL-2 and COX-2 were significantly decreased in CA-BSA-NPs-treated cells (FC  =  0.73 and 0.37, respectively) (Figure 11B,C) compared to control cells (*p* < 0.05 for each).

## 4. Discussion

The growth of and migration cancer cells of have been reported to be inhibited by carnosic acid [34]. Indeed, CA induces the ROS-mediated mitochondrial pathway to cause apoptosis in liver cancer cells [7]. However, the therapeutic efficacy of CA is limited due to low solubility and diminished bioavailability [35]. This work intended to develop a CA-nanoformula to improve its bioavailability and therapeutic activity. Also, the current study deals with the impact of CA on apoptosis via following the expression of one oncogene (GCLC) and two anti-apoptotic genes (BCL-2 and COX-2). In nanodrug delivery, the high EE%, LC%, and sustained in-vitro releasing are related to the increased medical value of the nano-formulated bioactive substances [36].

In this study, CA-CH-NPs and CA-BSA-NPs showed a comparable value of EE% and LC%. However, CA-BSA-NPs showed better release profiles over 10 h, which resulted in the release of 75% of encapsulated CA compared with 61% for CA-CH-NPs and 15.6% for CA-CL-NPs. So, the CA-BSA-NPs was chosen as the most promising formula and used in the following experiments.

The physicochemical properties of a nanocarrier, such as particle size, shape, and surface charge, play a crucial role in antitumor activity [37]. Indeed, the small particle size enhances nanoparticle penetration ability and retention in tumor tissue. Also, zeta potential is a valuable parameter for nanoparticles’ physical stability [38]. In the current study, TEM imaging indicates the spherical uniform size of both BSA-NPs and CA-BSA-NPs. The small PDI value confirmed the uniform size and successful preparation of BSA-NPs and CA-BSA-NPs. Moreover, the high negative value of zeta potential indicates the stability of the prepared nanomaterials.

The cell-based analysis showed a significant effect of CA-BSA-NPs on Caco-2 and MCF-7 cells. The IC_50_ of CA was significantly decreased by encapsulation in BSA-NPs. Interestingly, using the same concentration, both CA and empty carrier showed nonsignificant activity. Moreover, the considerable reduction in the cell viability after treatment with CA-BSA-NPs indicates that the encapsulation of CA in nanocarrier enhanced its cellular uptake and bioavailability. Similar results have been reported in the literature [39,40].

The molecular mechanisms triggering the cytotoxicity and apoptotic action of CA-BSA-NPs were investigated using cell cycle arrest analyses. The data showed the capability of CA-BSA-NPs to stimulate G2-phase cell cycle arrest on the treated cells [41,42].

Apoptosis is a characteristic mark of cell cytotoxicity [43]. In the current work, CA-BSA-NPs have been demonstrated to inhibit proliferation and promote apoptosis induction in the tested cells. Furthermore, the results revealed that CA-BSA-NPs treatment has a more noticeable effect on the Caco-2 than MCF-7 cells in the G2/M stage. This is in line with prior research, which found that CA inhibits DNA synthesis in Caco-2 cells and causes a brief cell cycle arrest in the G2/M phase. Treatment with CA-BSA-NPs appears to cause cell necroptosis in MCF-7 cells in a p21-dependent manner [31].

The induction of apoptosis via alterations of regulatory genes has become a focus of extensive research [44]. The downregulation of COX-2 and BCL-2 gene expression, in addition to the upregulation of GCLC gene expression, are main triggering factors for the cells to go through apoptosis [45]. In agreement with this fact, Caco-2 and MCF-7 cells treated with CA-BSA-NPs showed upregulated expression of the GCLC gene and downregulated expression of BCL-2 and COX-2 genes in Caco-2 and MCF-7 cells in comparison to control cells.

## 5. Conclusions

A novel composite (CA-BSA-NPs) was developed and evaluated for its antitumor activity against MCF-7 and Caco-2. The obtained results revealed the significance of using CA in nanoformulation, as indicated by comparing cell viability after exposure to free CA and CA-BSA-NPs.

Furthermore, cell cycle analysis showed the CA-BSA-NPs’ efficacy in triggering apoptosis and arresting cells in the G2/M phase, demonstrating the antiproliferative action of our formulation. Also, CA-BSA-NPs induced apoptosis in MCF-7 and Caco-2. RT-PCR-based gene expression analysis showed an upregulation of the GCLC gene and downregulation of the BCL-2 and COX-2 genes in the treated cells compared to control cells. In conclusion, this study concludes that CA-BSA-NPs represent an efficient composite to improve the biomedical activity of CA in treating colon and breast cancers.

## Figures and Tables

**Figure 1 molecules-27-04102-f001:**
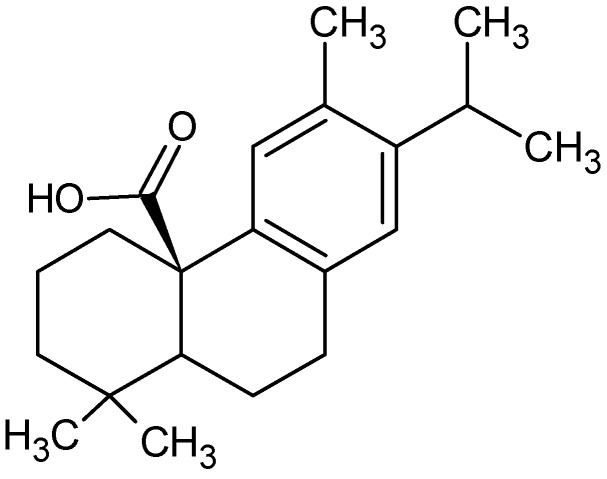
The chemical structure of carnosic acid.

**Figure 2 molecules-27-04102-f002:**
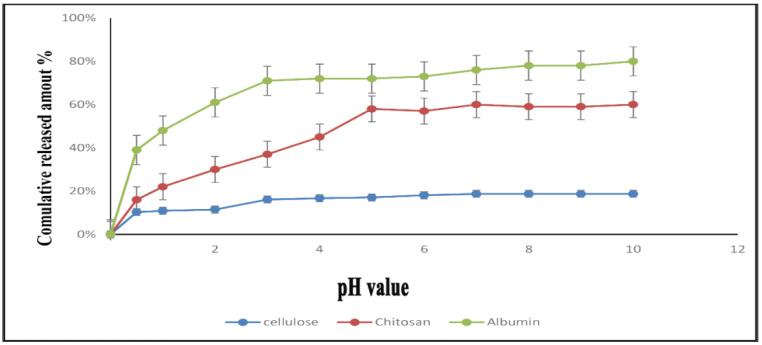
In vitro drug release study of carnosic acid from albumin, chitosan, and cellulose NPs at three pH values. Medium 1 (pH 1.5) for two hours, then medium 2 (pH 7.2) for 6 h, and finally medium 3 (pH 7.9) for 2 h.

**Figure 3 molecules-27-04102-f003:**
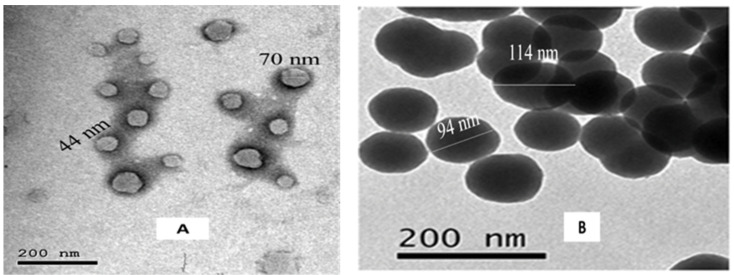
(**A**). TEM image of BSA-NPs. (**B**). TEM image of CA-BSA-NPs.

**Figure 4 molecules-27-04102-f004:**
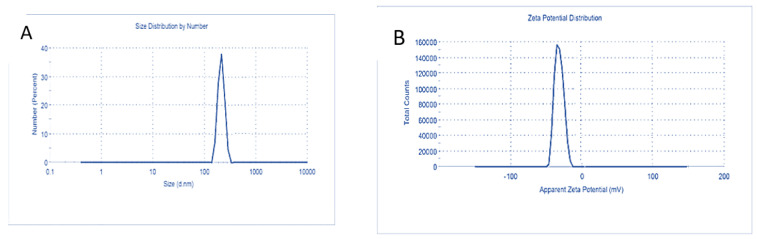
Characterization of BSA-NPs: (**A**). Particle size distribution BSA-NPs; (**B**). BSA-NPs zeta potential.

**Figure 5 molecules-27-04102-f005:**
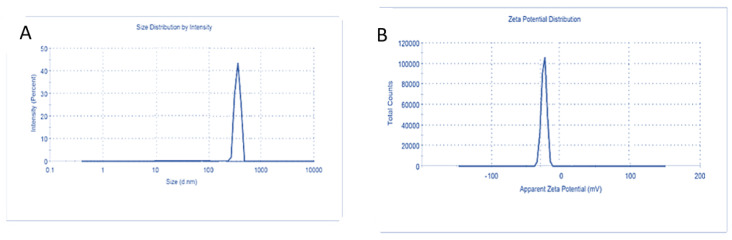
Characterization of CA-BSA-NPs: (**A**). Particle size distribution CA-BSA-NPs; (**B**). CA-BSA-NPs.zeta potential.

**Figure 6 molecules-27-04102-f006:**
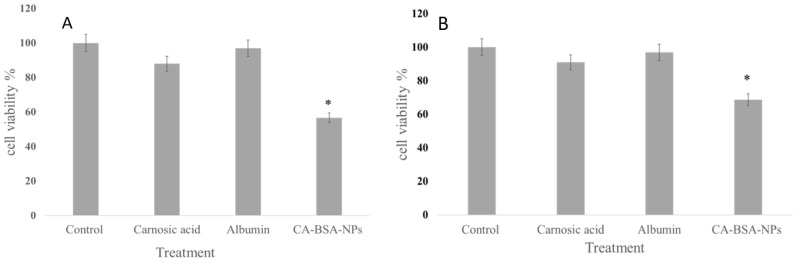
(**A**) Cell viability of Caco-2 cell line after treatment with carnosic acid, albumin and CA-BSA-NPs. (**B**). Cell viability of MCF-7 after treatment with carnosic acid, albumin and CA-BSA-NPs. * means significant difference *p* < 0.05.

**Figure 7 molecules-27-04102-f007:**
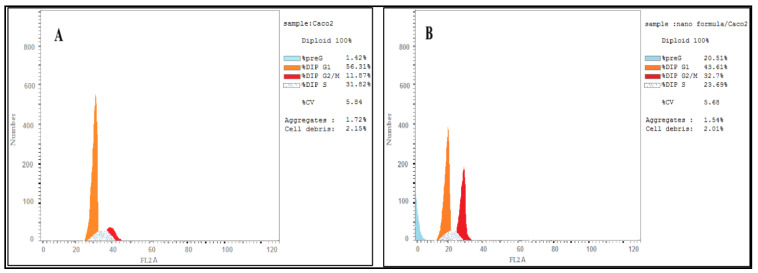
(**A**) Cell cycle analysis for Caco-2 cancer cell line (control); (**B**) cell cycle analysis for Caco-2 cancer cell line was treated with CA-BSA-NPs.

**Figure 8 molecules-27-04102-f008:**
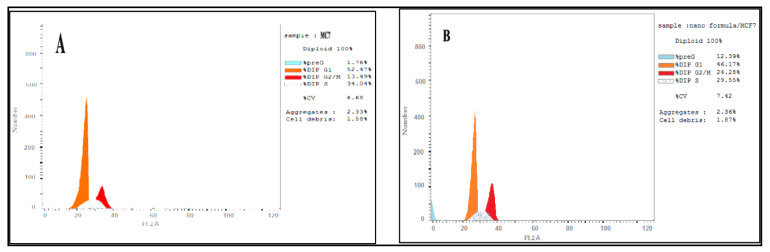
(**A**) Cell cycle analysis for MCF-7 cancer cell line (control). (**B**) Cell cycle analysis for MCF-7 cancer cell line was treated with CA-BSA-NPs.

**Figure 9 molecules-27-04102-f009:**
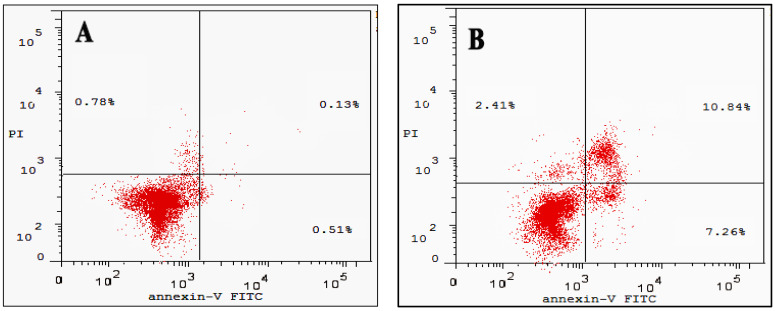
(**A**) Apoptosis analysis for Caco-2 (control); (**B**) Apoptosis analysis for Caco-2 cancer cell line was treated with CA-BSA-NPs.

**Figure 10 molecules-27-04102-f010:**
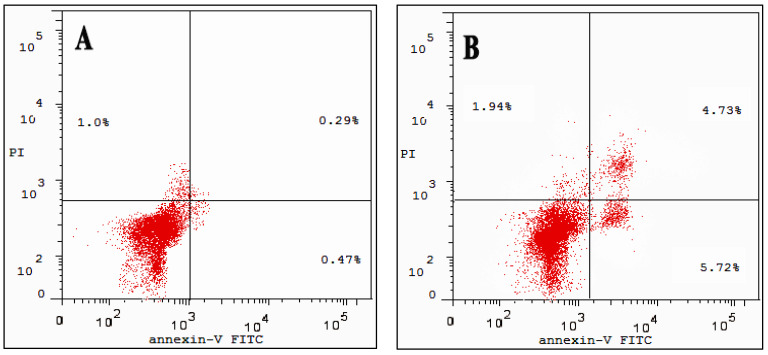
(**A**) Apoptosis analysis for MCF-7 (control); (**B**) apoptosis analysis for MCF-7 cancer cell line was treated with CA-BSA-NPs.

**Figure 11 molecules-27-04102-f011:**
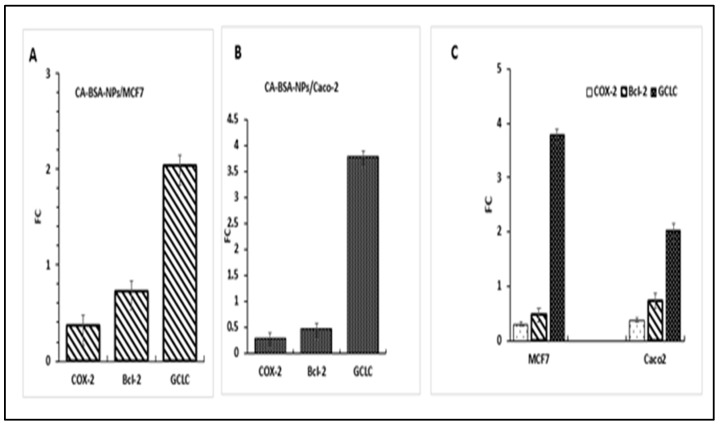
Gene expression profiling. (**A**): Fold change (FC) of GCLC, COX-2, and BCL-2 genes in MCF7cancer. (**B**): Fold change of GCLC, COX-2, and BCL-2 genes in Caco-2 cancer cells. (**C**): Fold change of GCLC, COX-2, and BCL-2 genes inMCF7 and Caco-2 cancer cells.

**Table 1 molecules-27-04102-t001:** The primer sequence of target genes and housekeeping gene.

Primer’s Name	Sequences of Primers
COX-2 forward COX-2 reverse	5′-GAATGGGGTGATGAGCAGTT-3′ 5′-CAGAAGGGCAGGATACAGC-3′
BCL-2 forward BCL-2 reverse	5′-CCTGTGGATGACTGAGTACC-3′ 5′-GAGACAGCCAGGAGAAATCA-3′
GCLC forward GCLC reverse	5′-GGCACAAGGACGTTCTCAAGT-3′ 5′-CAAAGGGTAGGATGGTTTGGG-3′
β-actin forward β-actin reverse	5′-GTGACATCCACACCCAGAGG-3′ 5′-ACAGGATGTCAAAACTGCCC-3′

## Data Availability

The data presented in this study are available.

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
