# Peer review of "Carnosic Acid Encapsulated in Albumin Nanoparticles Induces Apoptosis in Breast and Colorectal Cancer Cells"

_molecules, 2022, doi:10.3390/molecules27134102_

Round 1

Reviewer 1 Report

The paper is interesting, according to the requirements of the journal.
However, several inaccuracies occur.
First of all, more attention should be paid to how to write: equations (eg equation 1, page 4), subtitles, etc. English should be revised.
Page 6: I don't understand the dimensions of DLS. The value of 10292.8 (10 microns ?? !!) is very high, just like the PDI. Although the dimensions obtained at TEM are not compared with those at DLS, however, a magnitude of 100x as differences between the two characterization methods is far too large and must be seriously justified.
I do not understand figure 4; where do the 3 pHs that the authors are talking about appear on the figure?

Author Response

POINTWISE RESPONSE TO REVIEWERS’S COMMENTS ON Molecule -1738706

Dear Molecule Editor, Dear Reviewers,

Thank you for the reviewer comments for Molecule-1748097; they are very helpful to improve our manuscript. Please find attached the revised article entitled Carnosic Acid Encapsulated in Albumin Nanoparticles Induces Apoptosis in Breast and Colorectal Cancer cells, for consideration for publication in Molecule.

Each reviewer comment is indicated below, together with the responses (clarifications/ revisions) we have included in the revision.

It is hoped that this revision satisfies the queries raised by respected reviewers and manuscript will be considered for publication in Cancers.

With kind regards

Dr. Amr Hassan

Reviewer-1

The paper is interesting, according to the requirements of the journal. However, several inaccuracies occur.

  • First of all, more attention should be paid to how to write: equations (eg equation 1, page 4),

AUTHORS: Thank you very much for your valuable suggestion. Thanks for reviewer’s comments, all grammatical errors and topographical mistakes has been fixed. And the new edition of manuscript was revised carefully.

2- Page 6: I don't understand the dimensions of DLS. The value of 10292.8 (10 microns ?? !!) is very high, just like the PDI. Although the dimensions obtained at TEM are not compared with those at DLS, however, a magnitude of 100x as differences between the two characterization methods is far too large and must be seriously justified.

AUTHORS: Thank you very much for your valuable suggestion. Thanks for reviewer’s comments, the corrected DLS size value was added in the revised manuscript.

3- do not understand figure 6; where do the 3 pHs that the authors are talking about appear on the figure?

AUTHORS: Thank you very much for your valuable suggestion. Thanks for reviewer’s comments, The third pH has been corrected in the revised manuscript, also, Figure 6 has been corrected in the revised manuscript

 (Medium 1 (pH 1.5) for two hours, then medium 2 (pH 7.2) for 6 hours, and finally transferred to medium 3 (pH 7.9) for 2 hours). The three pHs were mentioned in the methodology with changing the medium in certain time

Reviewer 2 Report

The authors present the paper "Carnosic Acid Encapsulated with Bovine Serum Albumin Nanoparticles for Anti-Tumor Efficacy"

1) The fresh 2-3 year review papers are required for the Introduction section to present the area's importance and progress. I highly recommend to use mostly the review papers.

2) In the abstract you presented the meaning of breast cancer cells (MCF-7) and colon cancer cells (Caco-2) twice lines 21 and 26. You should present it only once.

3) Also, the references to the albumin is limited. It will be better to present a number of review papers. Moreover, you can mention that albumin increase biocompatibility, decrease immune response, increase solubility and colloidal stability, etc.

4) Figure 1 should be revised using Chem draw or other chemical formula programs. Section 2.2.14 present the Table title and I think it has to be COX-2 forward. The second line should be without spaces CCTGTG GAT GAC TGA GTA CC.

5) Can you present in a clear way the work's novelty in the Conclusion section, abstract, and the last part of the Introduction. The nanoparticle synthesis is general. Also, the limitation of the work is appropriate for the Conclusion section.

6) I highly recommend to present DLS size data, potential, and TEM results in a Table manner, too.

7) Section 3.2 is extremely short. It is difficult to understand about what compound capacity the text informed you. Also, it is the information about albumin and chitosan confuses: It indicated that nano-albumin is the better nanoparticle drug delivery system. But the loaded capacity is almost the same.

8) The section 3.4 Why you have used pH 1.5 where you can find it in the organism? Usually for the cell delivery, particles must have low drug release at pH 7.4 and high at ~5. How can you comment your experiment design? Also, in Figure 4: I don't see any three different pH values.

9) Figure 5 viability in percent??? But there are in the scale 0.36% in a control sample?

Author Response

POINTWISE RESPONSE TO REVIEWERS’S COMMENTS ON Molecule -1738706

Dear Microorganisms Editor, Dear Reviewers,

Thank you for the reviewer comments for Molecule-1738706; they are very helpful to improve our manuscript. Please find attached the revised article entitled Carnosic Acid Encapsulated in Albumin Nanoparticles Induces Apoptosis in Breast and Colorectal Cancer cells, for consideration for publication in Molecule.

Each reviewer comment is indicated below, together with the responses (clarifications/ revisions) we have included in the revision.

It is hoped that this revision satisfies the queries raised by respected reviewers and manuscript will be considered for publication in Cancers.

With kind regards

Dr. Amr Hassan

Reviewer-2

1) The fresh 2-3 year review papers are required for the Introduction section to present the area's importance and progress. I highly recommend using mostly the review papers.

AUTHORS: Thank you very much for your valuable suggestion. Thanks for reviewer’s comments, new citations were added in the revised edition, both in the introduction and discussion section.

2) In the abstract you presented the meaning of breast cancer cells (MCF-7) and colon cancer cells (Caco-2) twice lines 21 and 26. You should present it only once.

AUTHORS: Thank you very much for your valuable suggestion. Thanks for reviewer’s comments, the sentence of breast cancer cells (MCF-7) and colon cancer cells (Caco-2) was remove in the line 26. In the line26, we used the abbreviations MCF-7 and Caco-2.

3) Also, the references to the albumin are limited. It will be better to present a number of review papers. Moreover, you can mention that albumin increase biocompatibility, decrease immune response, increase solubility and colloidal stability, etc.

AUTHORS: Thank you very much for your valuable suggestion. Thanks for reviewer’s comments, new references that concerned about albumin were inserted in the introduction section.

4) Figure 1 should be revised using Chem draw or other chemical formula programs

AUTHORS: Thank you very much for your valuable suggestion. Thanks for reviewer’s comments; the figure 1 was draw by chem draw.

5) Section 2.2.14 presents the Table title and I think it has to be COX-2 forward. The second line should be without spaces CCTGTG GAT GAC TGA GTA CC

AUTHORS: Thank you very much for your valuable suggestion. Thanks for reviewer’s comments;  we remove the space in the revised manuscript.

6) Can you present in a clear way the work's novelty in the Conclusion section, abstract, and the last part of the Introduction? The nanoparticle synthesis is general. Also, the limitation of the work is appropriate for the Conclusion section.

 AUTHORS: Thank you very much for your valuable suggestion. Thanks for reviewer’s comments; we mention in the new edition the novelty of Encapsulation of Carnosic Acid with Albumin Nanoparticles which improve the solubility and bioavailability as well as increased the efficiency against numerous cancerous cell.

7) The section 3.4 Why you have used pH 1.5 where you can find it in the organism? Usually for the cell delivery, particles must have low drug release at pH 7.4 and high at ~5. How can you comment your experiment design? Also, in Figure 4: I don't see any three different pH values.

AUTHORS: Thank you very much for your valuable suggestion. Thanks for reviewer’s comments; According to the following references that study the methodology of varying the pH medium at 1.5, 7.2, and 7.9 which similar to Gastrointestinal tract pHs.

The references:

  1. Han, U., Seo, Y. and Hong, J., 2016. Effect of pH on the structure and drug release profiles of layer-by-layer assembled films containing polyelectrolyte, micelles, and graphene oxide. Scientific reports, 6(1), pp.1-10.
  2. Rasul, A., Khan, M.I., Rehman, M.U., Abbas, G., Aslam, N., Ahmad, S., Abbas, K., Shah, P.A., Iqbal, M., Al Subari, A.M.A. and Shaheer, T., 2020. In vitro characterization and release studies of combined nonionic surfactant-based vesicles for the prolonged delivery of an immunosuppressant model drug. International journal of nanomedicine, 15, p.7937.

8) Figure 5 viability in percent??? But there are in the scale 0.36% in a control sample?

AUTHORS: Thank you very much for your valuable suggestion. Thanks for reviewer’s comments; we draw the figure 5 and correct the mistake.

Reviewer 3 Report

It was a study about the synthesis and evaluation of the carnosic acid-loaded albumin nanoparticles for the aim of cancer therapy. Despite efforts done for the preparation of this study, it is not suitable to be published in this journal. Here are some comments on this study that could help to improve the quality of the manuscript:

  1. In the title, you mention just BSA as a drug carrier while in the abstract three different nanoparticles are mentioned as carriers.
  2. In the last paragraph of the introduction, you should describe the work you have done. What you mentioned for this is not complete. Please improve it.
  3. Please add the subheading “Materials” to the methods part and mention the materials used in the research, their purity, and the manufacturer company there.
  4. Please add references related to the preparation of nanoparticles.
  5. Please combine the drug-loaded NPs preparation with their preparation methods.
  6. “Carnosic acid encapsulation efficiency and loading capacity were performed as reported by Abdel-Hakeem et al., 2019” please add the reference.
  7. “The encapsulation efficiency of carnosic acid (EE) and carnosic acid (LC) loading capacity” please rewrite this sentence.
  8. “The data was analyzed using dynamic light scattering” what do you mean?
  9. “Three media re- 148 leases were prepared as follows: KCl/HCl buffer at pH 1.5 containing 30% ethanol (media 1), Phosphate buffer at pH 7.2 containing 30% ethanol (media 2), then ethanol was added to ensure sink conditions and free solubility of the released drug in the release media.” How about the third one? Why did you choose these three pHs? Did you check drug release without ethanol? Maybe its presence in release media affects the rate of drug release!
  10. “sonicating for 1 hour to extract all loaded drug” you couldn’t use sonication for drug release test.
  11. “Briefly, 1 × 105 cells/well were seeded in 96-well plates and exposed to CA, BSANPs, and CA-BSANPs at different concentrations.” why did you check just these ones? How about other nanoparticles?
  12. “. For the MTT solution (0.5 mg/ml), PBS (0.5 mg/ml) was added to the treated/washed plates (4 hr at a temperature of 37 ºC) as 10 μl per well until a purple-colored formazan complex was observed in the cytoplasm.” this sentence is not true. Please rewrite it.
  13. The quality of figure 1 is not good.
  14. “It indicated that nano-albumin is the better nanoparticulate drug delivery system.” according to the results chitosan acted better than others.
  15. “The mean diameter of BSANPs was 10292.8 nm and the polydispersity index (PDI) was 1.203.” these data are not acceptable and are not match with figure 5. Please repeat them.
  16. Please rewrite the figure captions of figures 3, 4, and 5.
  17. “with approximately 10.3% to 11.5% at 1.5 pH and 16.1 to 18.7% at 7.2 and 7.9” what do you mean?
  18. Figure 6 is contained just 1 pH of drug released results.
  19. Figure numbers are not corrected.
  20. Which concentrations did you use for the cell viability assay? Please add the results of all concentrations.
  21. Please rewrite the results of the apoptotic assay.
  22. There are some typos and grammatical mistakes in the text that should be corrected.
  23. Please use the same referencing method in the whole text.
  24. Please test the cytotoxicity effect of drug-loaded particles on a normal cell line, too.

Author Response

POINTWISE RESPONSE TO REVIEWERS’S COMMENTS ON Molecule -1738706

Dear Microorganisms Editor, Dear Reviewers,

Thank you for the reviewer comments for Molecule-1738706; they are very helpful to improve our manuscript. Please find attached the revised article entitled Carnosic Acid Encapsulated in Albumin Nanoparticles Induces Apoptosis in Breast and Colorectal Cancer cells, for consideration for publication in Molecule.

Each reviewer comment is indicated below, together with the responses (clarifications/ revisions) we have included in the revision.

It is hoped that this revision satisfies the queries raised by respected reviewers and manuscript will be considered for publication in Cancers.

With kind regards

Dr. Amr Hassan

Reviewer-3

It was a study about the synthesis and evaluation of the carnosic acid-loaded albumin nanoparticles for the aim of cancer therapy. Despite efforts done for the preparation of this study, it is not suitable to be published in this journal. Here are some comments on this study that could help to improve the quality of the manuscript:

  • In the title, you mention just BSA as a drug carrier while in the abstract three different nanoparticles is mentioned as carriers.

AUTHORS: Thank you very much for your valuable suggestion. Thanks for reviewer’s comments, we corrected in the abreact section

2- In the last paragraph of the introduction, you should describe the work you have done. What you mentioned for this is not complete. Please improve it.

AUTHORS: Thank you very much for your valuable suggestion. Thanks for reviewer’s comments; we improved the last paragraph in the introduction “For this purpose, carnosic acid was encapsulated in different polymeric nanoparti-cles named chitosan (CH), bovine serum albumin (BSA), and cellulose (CL). The prepared nanoformulations were characterized to select the best formula. The selected formula was utilized as a treatment for MCF-7 and Caco-2. The antitumor activity was followed via MTT assay and cell cycle analysis. Moreover, GCLC, BCL-2, and COX-2 gene expressions were also evaluated before and after treatment.’ 

Please add references related to the preparation of nanoparticles

AUTHORS: Thank you very much for your valuable suggestion. Thanks for reviewer’s comments, we inserted  references that concerned about  the preparation of nanoparticles in material and method section.

4) Carnosic acid encapsulation efficiency and loading capacity were performed as reported by Abdel-Hakeem et al., 2019” please add the reference

AUTHORS: Thank you very much for your valuable suggestion. Thanks for reviewer’s comments; we added Abdel-Hakeem et al., 2019 reference “Abdel-Hakeem, M.A., et al., Doxorubicin loaded on chitosan-protamine nanoparticles triggers apoptosis via downregulating Bcl-2 in breast cancer cells. Journal of Drug Delivery Science and Technology, 2020. 55: p. 101423.”

5) “The encapsulation efficiency of carnosic acid (EE) and carnosic acid (LC) loading capacity” please rewrite this sentence.

AUTHORS: Thank you very much for your valuable suggestion. Thanks for reviewer’s comments;  we rewrite this sentence in the revised manuscript.

6) “The data was analyzed using dynamic light scattering” what do you mean?  

AUTHORS: Thank you very much for your valuable suggestion. Thanks for reviewer’s comments; hydrodynamic size and zeta potential were measured by dynamic light scattering.

7) “Three media re- 148 leases were prepared as follows: KCl/HCl buffer at pH 1.5 containing 30% ethanol (media 1), Phosphate buffer at pH 7.2 containing 30% ethanol (media 2), then ethanol was added to ensure sink conditions and free solubility of the released drug in the release media.”

 How about the third one?

Why did you choose these three pHs?

AUTHORS: Thank you very much for your valuable suggestion. Thanks for reviewer’s comments; According to the following references that study the methodology of varying the pH medium at 1.5, 7.2, and 7.9 which similar to Gastrointestinal tract pHs.

The references:

  1. Han, U., Seo, Y. and Hong, J., 2016. Effect of pH on the structure and drug release profiles of layer-by-layer assembled films containing polyelectrolyte, micelles, and graphene oxide. Scientific reports, 6(1), pp.1-10.
  2. Rasul, A., Khan, M.I., Rehman, M.U., Abbas, G., Aslam, N., Ahmad, S., Abbas, K., Shah, P.A., Iqbal, M., Al Subari, A.M.A. and Shaheer, T., 2020. In vitro characterization and release studies of combined nonionic surfactant-based vesicles for the prolonged delivery of an immunosuppressant model drug. International journal of nanomedicine, 15, p.7937.

8) Did you check drug release without ethanol? Maybe its presence in release media affects the rate of drug release!

AUTHORS: Thank you very much for your valuable suggestion. Thanks for reviewer’s comments; Ethanol was added to ensure sink conditions and free solubility of released drug in the release media due to the poor solubility of carnosic acid in the tested media.

9)“ 13.   The quality of figure 1 is not good

AUTHORS: Thank you very much for your valuable suggestion. Thanks for reviewer’s comments; we draw the figure one by chem draw.

The mean diameter of BSANPs was 10292.8 nm and the polydispersity index (PDI) was 1.203.” these data are not acceptable and are not match with figure 5. Please repeat them .

AUTHORS: Thank you very much for your valuable suggestion. Thanks for reviewer’s comments;  we repeated the experimental and results were” CA-BSA-NPs had a hydrodynamic size range of 330 to 662 nm after CA loading, with a primary peak at 520 nm, polydispersity index (PDI) of 0.119, and zeta potential of -21.03mV (Fig. 5 A, B)”.

Please add the subheading “Materials” to the methods part and mention the materials used in the research, their purity, and the manufacturer company there.

AUTHORS: Thank you very much for your valuable suggestion. Thanks for reviewer’s comments; 

               Materials:

Breast cancer cells (MCF-7) and colorectal cancer (Caco-2) were purchased from VACSERA, Giza, Egypt. Carnosic acid standard (purity 99%), chitosan (Deacetylation degree 95%, molecular weight 80 kDa), and BSA were purchased from Sigma Aldrich. Ethyl acetate and hexane were purchased from Fisher scientific. Sodium tripolyphosphate (TPP) and glutaraldehyde were purchased from Merck. All the other chemicals and reagents were analytical grade.

Please rewrite the figure captions of figures 3, 4, and 5.

AUTHORS: Thank you very much for your valuable suggestion. Thanks for reviewer’s comments;   we corrected the figures caption.

Figure 6 is contained just 1 pH of drug released results.

AUTHORS: Thank you very much for your valuable suggestion. Thanks for reviewer’s comments;  

The required corrections have been added to the revised manuscript “(Medium 1 (pH 1.5) for two hours, then medium 2 (pH 7.2) for 6 hours, and finally transferred to medium 3 (pH 7.9) for 2 hours.)

Figure numbers are not corrected

AUTHORS: Thank you very much for your valuable suggestion. Thanks for reviewer’s comments;  

The figures numbers were corrected

Which concentrations did you use for the cell viability assay? Please add the results of all concentrations.

AUTHORS: Thank you very much for your valuable suggestion. Thanks for reviewer’s comments;  

The IC50 values of free CA against Caco-2 and MCF-7 cells were 8.29 mg/ml and 27.43 mg/ml, respectively. Interestingly, the CA-BSA-NPs significantly reduced the IC50 values to 2.60 and 6.02 μg/ml for Caco-2 and MCF-7, respectively. Furthermore, after treatment with the same concentration, the MTT assay showed a significant decrease in the cell viability after treatment with CA-BSA-NPs, while the BSA-NPs and free CA showed nonsignificant cytotoxicity (Fig. 6 A, B).

Please rewrite the results of the apoptotic assay

AUTHORS: Thank you very much for your valuable suggestion. Thanks for reviewer’s comments;   we rewrite the apoptotic assay “Annexin V-FITC/PI staining was used to identify apoptosis in the treated and un-treated cancer cell lines. Data in Fig. 9a showed that total apoptosis (undergone early and late apoptosis) in untreated cells was 0.64%, while CA-BSA-NPs-treated Caco-2 cancer cells yielded 17.74% (figure 9b). On the other hand, CA-BSA-NPs had a lower effect on MCF-7 cells. As Figure 10b displayed, the percentages of apoptotic cells (including early and late apoptotic cells) in MCF-7 treated cells were increased to 10.45% compared with control (1.05%) in Figure 10a.”

There are some typos and grammatical mistakes in the text that should be corrected.

AUTHORS: Thank you very much for your valuable suggestion. Thanks for reviewer’s comments;  all the grammatical mistakes were fixed.

Please use the same referencing method in the whole text.

AUTHORS: Thank you very much for your valuable suggestion. Thanks for reviewer’s comments; we unity the referencing method in the whole text [  ].

Please test the cytotoxicity effect of drug-loaded particles on a normal cell line, too.

AUTHORS: Thank you very much for your valuable suggestion. Thanks for reviewer’s comments; we completely agree with the reviewer regarding the cytotoxicity effect point. The cells and nano formulations are not available yet to perform this test. At the same time, we also try to get a suitable fund to do some experiments related to cell viability using MTT assay or Annex V but unfortunately, we cannot get this fund till now. But we appreciate your support in getting this work published to support our status in the project that we applied for to have an excellent chance to prepare our project. However, different literatures don’t correlate the effect of the nanoparticles on normal and tumor cells. 

Round 2

Reviewer 1 Report

I accept publication taking into account the new modification.

Author Response

POINTWISE RESPONSE TO REVIEWERS’S COMMENTS ON Molecule -1738706

Dear Molecule Editor, Dear Reviewers,

Thank you for the reviewer comments for Molecule-1748097; they are very helpful to improve our manuscript. Please find attached the revised article entitled Carnosic Acid Encapsulated in Albumin Nanoparticles Induces Apoptosis in Breast and Colorectal Cancer cells, for consideration for publication in Molecule.

Each reviewer comment is indicated below, together with the responses (clarifications/ revisions) we have included in the revision.

It is hoped that this revision satisfies the queries raised by respected reviewers and manuscript will be considered for publication in Cancers.

With kind regards

Dr. Amr Hassan

Reviewer-1

I accept publication taking into account the new modification

AUTHORS: Thank you very much for your valuable suggestion. Thanks for reviewer’s comments, we addition a new paragraphs about the role of albumin “Albumin is an attractive macromolecule carrier that can be obtained from a various sources including egg white (ovalbumin), bovine serum albumin, and human serum albumin (HSA). ). It is a water soluble protein that maintains the osmotic pressure, binding, and transport of nutrients to the cells [17].  It can be used as an eco-friendly biomaterial in drug delivery because of its biocompatibility and easy degradation without toxicity. Bovine serum albumin (BSA) micelles have been developed to improve the bioavailability of these drugs and reduce their toxicity [18]. Histological observations demonstrated that bovine albumin has no adverse effects following frequent administration by the intranasal route [19]. Additionally, the application of BSA as a drug delivery was showed in MCF-7 xenograft mouse model where the in vivo antitumor evaluation of FA-Rg5-BSA NPs showed more effective in inhibiting tumor growth than Rg5 [20-21].”

Reviewer 2 Report

Dear authors, thank you for the revised paper. However, I have some minor suggestions.

In the previous version you have the information about albumin advantages of albumin in the paper Introduction section. I think several sentences about albumin advantage should be included with several review references. The paper is called "Carnosic Acid Encapsulated in Albumin Nanoparticles Induces Apoptosis in Breast and Colorectal Cancer cells". In this way, not to present some information about albumin looks weird. 

Fig 2 -20% have to be deleted. I recommend writing on the X-scale that it is the pH values.

Author Response

POINTWISE RESPONSE TO REVIEWERS’S COMMENTS ON Molecule -1738706

Dear Microorganisms Editor, Dear Reviewers,

Thank you for the reviewer comments for Molecule-1738706; they are very helpful to improve our manuscript. Please find attached the revised article entitled Carnosic Acid Encapsulated in Albumin Nanoparticles Induces Apoptosis in Breast and Colorectal Cancer cells, for consideration for publication in Molecule.

Each reviewer comment is indicated below, together with the responses (clarifications/ revisions) we have included in the revision.

It is hoped that this revision satisfies the queries raised by respected reviewers and manuscript will be considered for publication in Cancers.

With kind regards

Dr. Amr Hassan

Reviewer-2

1) In the previous version you have the information about albumin advantages of albumin in the paper Introduction section. I think several sentences about albumin advantage should be included with several review references. The paper is called "Carnosic Acid Encapsulated in Albumin Nanoparticles Induces Apoptosis in Breast and Colorectal Cancer cells". In this way, not to present some information about albumin looks weird. 

AUTHORS: Thank you very much for your valuable suggestion. Thanks for reviewer’s comments, we addition a new paragraphs about the role of albumin “Albumin is an attractive macromolecule carrier that can be obtained from a various sources including egg white (ovalbumin), bovine serum albumin, and human serum albumin (HSA). It is a water soluble protein that maintains the osmotic pressure, binding, and transport of nutrients to the cells [17].  It can be used as an eco-friendly biomaterial in drug delivery because of its biocompatibility and easy degradation without toxicity. Bovine serum albumin (BSA) micelles have been developed to improve the bioavailability of these drugs and reduce their toxicity [18]. Histological observations demonstrated that bovine albumin has no adverse effects following frequent administration by the intranasal route [19]. Additionally, the application of BSA as a drug delivery was showed in MCF-7 xenograft mouse model where the in vivo antitumor evaluation of FA-Rg5-BSA NPs showed more effective in inhibiting tumor growth than Rg5 [20-21].”

2) Fig 2 -20% have to be deleted. I recommend writing on the X-scale that it is the pH values

 AUTHORS: Thank you very much for your valuable suggestion. Thanks for reviewer’s comments, we deleted figure -20%. Also, we rewrite pH instead of time as you suggested

Reviewer 3 Report

1- "Briefly, In 96-well plates, 1 × 105 cells/well were seeded and exposed to CA, BSA-NPs, and 176 CA-BSA-NPs at different concentrations." The authors didn't mention anything about these concentrations. Figure 6 and its figure caption should be corrected.

2- "Ten μl of DMSO". It is not correct, normally 100 μl of DMSO was added for 10 μl of MTT. 

3- The quality of Figures 9-b and 10-b should be improved.

4- what do you conclude from the results of the apoptotic assay? 

Author Response

POINTWISE RESPONSE TO REVIEWERS’S COMMENTS ON Molecule -1738706

Dear Microorganisms Editor, Dear Reviewers,

Thank you for the reviewer comments for Molecule-1738706; they are very helpful to improve our manuscript. Please find attached the revised article entitled Carnosic Acid Encapsulated in Albumin Nanoparticles Induces Apoptosis in Breast and Colorectal Cancer cells, for consideration for publication in Molecule.

Each reviewer comment is indicated below, together with the responses (clarifications/ revisions) we have included in the revision.

It is hoped that this revision satisfies the queries raised by respected reviewers and manuscript will be considered for publication in Cancers.

With kind regards

Dr. Amr Hassan

Reviewer-3

It was a study about the synthesis and evaluation of the carnosic acid-loaded albumin nanoparticles for the aim of cancer therapy. Despite efforts done for the preparation of this study, it is not suitable to be published in this journal. Here are some comments on this study that could help to improve the quality of the manuscript:

Briefly, In 96-well plates, 1 × 105 cells/well were seeded and exposed to CA, BSA-NPs, and 176 CA-BSA-NPs at different concentrations." The authors didn't mention anything about these concentrations. Figure 6 and its figure caption should be corrected.

AUTHORS: Thank you very much for your valuable suggestion. Thanks for reviewer’s comments, we mentioned the concentration which used in the experiment “Briefly, In 96-well plates, 1 × 105 cells/well were seeded and exposed to CA, BSA-NPs, and CA-BSA-NPs at different concentrations(100μg/ml ,50μg/ml, 25μg/ml , 12.5μg/ml and 6.25 μg/ml).”

2- Ten μl of DMSO". It is not correct, normally 100 μl of DMSO was added for 10 μl of MTT

AUTHORS: Thank you very much for your valuable suggestion. Thanks for reviewer’s comments; we corrected the sentence to be 100 μl of DMSO was added for10μl of MTT.

3) The quality of Figures 9-b and 10-b should be improved

AUTHORS: Thank you very much for your valuable suggestion. Thanks for reviewer’s comments; we improved figures 9 a, 9b, 10 a and 10b.

4) What do you conclude from the results of the apoptotic assay?

AUTHORS: Thank you very much for your valuable suggestion. Thanks for reviewer’s comments;  as a result, the data displayed that CA-BSA-NPs induce apoptosis on both MCF-7 and Caco-2 cell lines.
